∂ | **Open Peer Review** | Host-Microbial Interactions | Research Article

# Microbial transfer through fecal strings on eggs affects leaf beetle microbiome dynamics

Yueqing An,[1] Sarahi L. Garcia,[1,2] Peter A. Hambäck[1]

**ABSTRACT** Gut microbiomes of holometabolous insects can be strongly affected by metamorphosis. Previous studies suggest that microbiome colonization and community development often rely on specialized transmission routes between host life stages. However, there is a lack of comparative studies of microbial community dynamics from different transmission mechanisms. We compared the gut microbial community dynamics across life stages in five *Galerucella* species that differ in their potential microbial transfer mechanism by sequencing amplicons of the 16S rRNA gene. Females of three of the studied species place a fecal string on top of the egg, which may enhance the transfer of gut microbes, whereas females of the two other species do not. We found that the α-diversity was more stable between life stages in fecal string-placer species compared with the non-fecal string-placer species. Moreover, there were consistent microbiome differences between species, with multiple taxa in each species consistently appearing in all life stages. Fecal strings placed on eggs seem to play an important role in the diversity and dynamics of gut bacteria in *Galerucella* species, facilitating the vertical transfer of gut bacteria between host insect generations. Alternative, but less efficient, transmission routes appear to occur in non-fecal string-placer species.

**IMPORTANCE** We explore the consequences of having different mechanisms for transferring and establishing the gut microbiome between generations on gut microbial community dynamics. This process is often problematic in holometabolous insects, which have a complete metamorphosis between larval and adult stages. In our previous research, we found that females of some species within the genus *Galerucella* (Chrysomelidae) place a fecal string on the eggs, which is later consumed by the hatching larvae, whereas other species in the same genus do not have this behavior. In this paper, we therefore quantify the microbial community dynamics across all life stages in five *Galerucella* beetles (three with and two without fecal strings). Our results also indicate that the dynamics are much more stable in the species with fecal strings, particularly in the early life stages.

**KEYWORDS** gut microbiome, microbial ecology, 16S RNA

Many insects have evolved close and complex relationships with microorganisms, and this symbiosis provides several benefits for the insect host (1). Microorganisms residing in the insect gut may provide key nutrients and help sequester plant defenses (2), but they may also be involved in defending against natural enemies (3). Not surprisingly, some insect species have evolved elaborate mechanisms for transferring microorganisms from mother to offspring. However, such vertical transmission is not the only source of gut microbes, as these can also be derived from food, the environment, or acquired through social interactions (horizontal transmission) (4). This diversity of mechanisms results in the gut microbiome being a mixture of microorganisms picked up from the environment and those transferred across generations. Accordingly, we should

Address correspondence to Yueqing An, yueqing.an1994@gmail.com.

The authors declare no conflict of interest.

See the funding table on p. 11.

expect both host phylogeny and environmental factors (such as host diet or gut pH) to shape gut microbial communities (5).

The different sources for gut microbes, whether they are vertically transmitted or picked up from the environment, are likely to affect microbial community dynamics across insect life stages (6, 7). The life history of holometabolous insects, which undergo complete metamorphosis between larval and adult stages, includes stages where transitions between life stages are particularly complicated. First, most insect larvae have little contact with their mother, making it difficult to actively transfer gut microbes to their offspring. Similarly, microbial transfers are further complicated by the pupal stage, during which the insect gut is often completely restructured. Some insects have overcome these challenges through specialized storage spaces in eggs or pupae, allowing microorganisms to be efficiently transferred between generations (6–8). Other insects rely on the immune system to manage microbes acquired from the environment (9). While the number of studies documenting differences between species and/or life stages is increasing—showing more or less strong phylogenetic signals in gut microbial communities (10–13)—there is still a lack of comparative studies on how different transfer mechanisms affect microbial community dynamics across insect life stages.

In our previous work, we discovered a set of closely related beetle species (Chrysomelidae: *Galerucella*) that differ in a potential mechanism for transferring gut microbial communities (14). In this genus, females of some species place a fecal string on top of their eggs (Fig. 1), while females of other species do not (hereafter referred to as fecal and non-fecal species, respectively). Fecal strings may serve multiple functions, but one possible role is to provide newly hatched larvae with the female's microbial community. It would, therefore, be of interest to compare the microbial community dynamics between species that do or do not produce such fecal strings. One would expect species with an efficient transfer mechanism to have a more stable microbial community, particularly immediately after transfer, compared to species that must re-establish their community each generation from environmentally acquired microbes.

## MATERIALS AND METHODS

### Sample collection

To track the microbial composition, we collected adults of five *Galerucella* species by hand in the field during early June. The specimens were collected in small vials together with pieces of host plant leaves. The species include three species [Gl—*G. lineola* (Fabricius, 1781), Gp—*G. pusilla* (Duftschmid, 1825), and Gc—*G. calmariensis* (L., 1767)] that place a fecal string on the egg (Fig. 1) and two species [Gn—*G. nymphaea* (L., 1758) and Gs—*G. sagittariae* (Gyllenhal, 1813)] that do not place such fecal strings. In the area, four species are functionally monophagous, where they typically feed on only one host plant species in each locality. Three of these species have only one host plant (Gp—*Lythrum salicaria*, Gc—*Lythrum salicaria*, and Gn—*Nuphar lutea*), and one species (Gs) has three host plants (*Lysimachia vulgaris*, *L. thyrsiflora*, and *Comarum palustre*) in different sites and was collected on *L. vulgaris*. Finally, the host spectrum of Gl includes a range of *Salix* species. The species are all holometabolous, meaning that they have a life cycle where the transition between larvae and adults includes a pupal stage, and the timing of life stages is similar. Other factors, such as habitat, are also very similar among species, with the exception of Gn that feeds on plants out in the water.

In the lab, we placed adults in cages ($30 \times 30 \times 60$ m$^3$) on potted plants of the same species as collected in the field, with the exception of Gl. For this species, we, for practical reasons, used *Salix viminalis*. Thus, the host plants used in the trials included *S. viminalis* (Gl), *L. salicaria* (Gp and Gc), *N, lutea* (Gn), and *L. vulgaris* (Gs). The different beetle species were isolated from one another in plastic cages within the same laboratory space and under identical environmental conditions (about 21°C), including management and diet. In the cages, adults immediately laid eggs on their host plant. The eggs were transferred

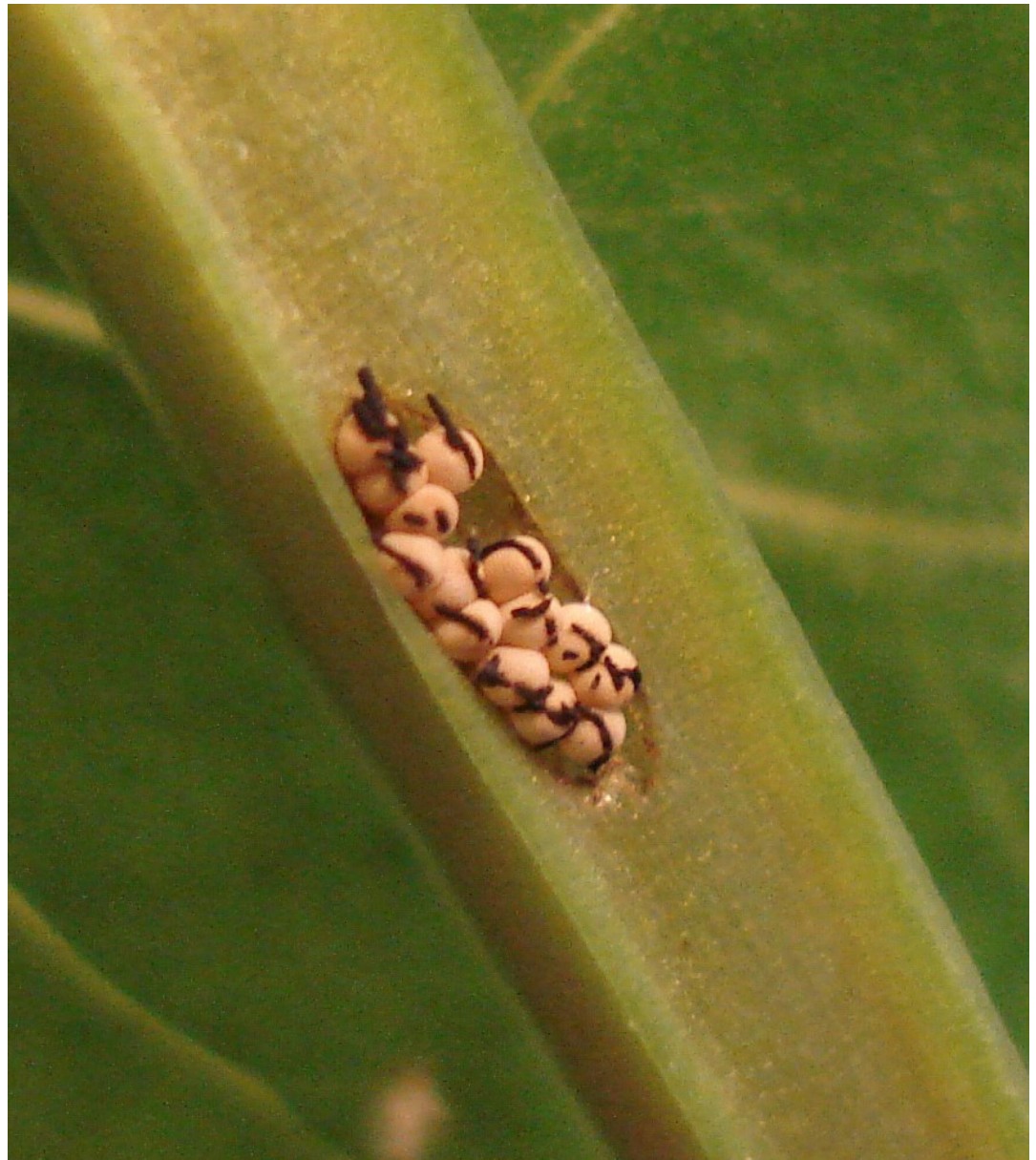

**FIG 1** Egg from *Galerucella calmariensis*, with fecal string. Photo: Robert Markus.

to petri dishes and hatched naturally in a few weeks. We collected samples for sequencing from five life stages: eggs, newly hatched larvae, 3-week-old larvae, pupae, and newly emerged adults (second generation). Three-week-old larvae are nearly fully developed and would pupate within a few days. The reason for using time as a factor for selecting larvae instead of larval stages was that these stages are difficult to identify in these species. A total of 172 samples were obtained and kept in 95% ethanol at 4°C until further processing (Table 1).

## Sample preparation and DNA extraction

DNA extraction was performed using the DNeasy Blood and Tissue Kit (Qiagen, cat. no. 69 504). A negative control with water instead of DNA (control for contamination) was included. The extracted DNA was stored at −20°C until amplicons were generated. The V3–V4 hypervariable regions (480 bp) of the 16S rRNA gene were amplified using the universal primer pair 341F (5′-ACACTCTTTCCCTACACGACGCTCTTCCGATCT-3′) and 805R

**TABLE 1** Sample information of five *Galerucella* species

| Beetle species | Host plant | Life stage | # of individuals | Sequences per sample |
|---|---|---|---|---|
| *G. calmariensis* | *L. salicaria* | Eggs | 10 | 17,303.5 |
| | | Newly hatched larvae | 10 | 22,380.7 |
| | | Three-week old larvae | 10 | 11,767.2 |
| | | Newly hatched adults | 9 | 34,073.22 |
| *G. pusilla* | *L. salicaria* | Newly hatched larvae | 10 | 41,303.9 |
| | | Three-week old larvae | 5 | 15,296.4 |
| | | Pupae | 10 | 60,672.2 |
| *G. lineola* | *Salix* spp. | Eggs | 10 | 64,686.4 |
| | | Newly hatched larvae | 10 | 69,087.2 |
| | | Three-week old larvae | 6 | 32,115.67 |
| *G. sagittariae* | *L. vulgaris* | Newly hatched larvae | 15 | 62,025.73 |
| | | Three-week old larvae | 8 | 78,627 |
| | | Pupae | 9 | 48,388 |
| | | Newly hatched adults | 10 | 17,045.6 |
| *G. nymphaea* | *N. alba* | Eggs | 10 | 49,714.3 |
| | | Newly hatched larvae | 10 | 35,774.9 |
| | | Pupae | 10 | 57,006.3 |
| | | Newly hatched adults | 10 | 61,137.6 |

(5′-GTGACTGGAGTTCAGACGTGTGCTCTTCCGATC-3′) attached to Illumina adapters (15). The PCR program used KAPA HiFi HotStart Ready Mix (cat. no. 07958927001, Roche): 98°C for 2 min, followed by 35 cycles of 98°C for 20 s, 49°C for 15 s, and 72°C for 2 min. After measuring and normalizing the concentration of PCR products using the Qubit dsDNA HS Assay Kit, the products were sequenced by the Swedish National Genomics Infrastructure (NGI). The NGI amplicon sequencing setup included two cleanup steps, PCR indexing, and the final library preparation. PCR purification used MagSI beads and paired-end sequencing (2 × 300 bp) were performed on a MiSeq3 platform.

## Bioinformatic and statistical analysis

The quality of raw sequences was checked using FastQC and filtered using a phred-score cutoff = 20 before merging forward and reverse strands. Raw reads were trimmed to the fragment length (480 bp) by removing adaptors in R 4.1.1 (16). We further processed and filtered sequences, excluded chimeras, and clustered them into the amplicon sequence variants (ASVs) using DADA2 (17). ASVs restricted to single samples were also excluded. To better understand the differences in the microbiome between fecal and non-fecal species at different life stages, ASV sequences were compared against the annotated SILVA 138 SSU 16S rRNA reference database (https://www.arb-silva.de/) (18) using BLAST. After removing ASVs of chloroplasts and mitochondria, the remaining ASV data were summarized and compared at phylum and genus levels for subsequent analyses. Relative ASV abundances were analyzed across the five beetle life stages and compared between species. The α-diversity was measured using the abundance-based coverage estimators (ACE) index, a diversity metric that defines $S_{abun}$ as the number of abundant taxa and $S_{rare}$ as the number of rare taxa. We estimated β-diversity of the microbial communities between fecal and non-fecal species using both principal coordinates analysis (PCoA) based on the Bray-Curtis index and a permutational multivariate analysis of variance. Venn diagrams were visualized by the online tool Interactive Venn (http://www.interactivenn.net). To construct co-occurrence networks for each developmental stage, the Pearson correlation coefficient algorithm, based on relative bacterial abundance, was used to infer co-occurrence patterns (15). Significant correlations between two genera were defined as $0.25 < r < −0.25$ with an FDR-adjusted $P$-value $< 0.05$. For the analyses, we used MicroBiomeAnalyst 2.0 (19).

## RESULTS

### 16S rRNA gene sequencing analysis and taxa generated

A total of 7,564,344 sequences were obtained from 172 samples across the five beetle species and the various life stages (Table 1), with 282 to 106,349 sequences per sample (mean = 43,724). These sequences were clustered into 9,986 ASVs. When blasting ASV sequences against the SILVA 16S rRNA database, we found 26 bacterial phyla. However, four dominant phyla accounted for 99.62% of all sequences, so the remaining ones will not be further discussed. Ranked by relative abundance, dominant phyla were Proteobacteria (86.10%), Actinobacteria (4.65%), Firmicutes (3.82%), and Bacteroidetes (5.05%) (Fig. S1).

The gut microbiome of each life stage and species was dominated by a set of core bacteria. *Pseudomonas* was the most abundant genus in both fecal and non-fecal species, though its frequency decreased toward the pupal and adult stages (Fig. 2). Other common genera in both fecal and non-fecal species, but with varying frequencies, included *Stenotrophomonas*, *Pedobacter*, *Acinetobacter*, and *Serratia*, each representing more than 3% of the total bacteria in the entire data set. Non-fecal species had a particularly high fraction of *Serratia* in larvae, pupae, and adults, while *Wolbachia* and *Flavobacterium* were found almost exclusively in non-fecal species, especially in the larval and adult stages. Fecal species had a high dominance of *Pseudomonas* at egg, larvae, and adult stages, but *Acinetobacter* was also a dominant genus in newly hatched larvae of these species.

### Variation of microbial diversity among five life stages of *Galerucella*

To evaluate the role of insect fecal strings in shaping microbial community dynamics across life stages, we examined differences in community diversity between species with and without fecal strings (Fig. 3). The α-diversity of the gut microbial community was higher in non-fecal species during adult, egg, and larval stages and highest in fecal species during the pupal stage. At the same time, the variation in α-diversity between life stages was lower in fecal species compared to non-fecal species.

The ASV profiles differed between fecal and non-fecal species and across development stages, indicating that the microbiota is dynamic during growth rather than synchronized (Fig. 4; Fig. S3). The PCoA revealed a separation of samples between life stages within both fecal ($F = 4.14$, $R = 0.16$, $P = 0.001$) and non-fecal species ($F = 4.19$, $R = 0.18$, $P = 0.001$) (Fig. S3). In the PCoA, the first and second principal components (PC1 and PC2) explained 18.4 and 12%, respectively, of the variation in the microbial community for fecal species and 15.8 and 9.1%, respectively, for non-fecal species. Thirty ASVs were shared across all life stages in fecal species, including *Novosphingobium* and *Flavobacterium* (Fig. 4a). Non-fecal species shared 50 ASVs across all life stages, including the genera *Acinetobacter*, *Erwinia*, and *Wolbachia* (Fig. 4b). Some ASVs also overlapped across all life stages for both fecal and non-fecal species, including species within *Pseudomonas* and *Stenotrophomonas* (Fig. 4c).

### Host fecal string shapes the gut microbiota throughout life

As bacteria interactions are key modulators shaping the gut microbiota, we used bacterial co-occurrence network analysis to evaluate the role of fecal strings on bacteria interactions across the different life stages (Fig. 5). These analyses indicated differences in the network structure with hub bacteria between fecal and non-fecal species. Hub bacteria included *Pseudomonas*, *Stenotrophomonas*, *Pedobacter*, *Sphingobacterium*, *Acinetobacter*, *Serratia*, *Erwinia*, *Novosphingobium*, *Flavobacterium*, and *Wolbachia* (Fig. 5). These 10 bacteria exhibited varying relative abundances between fecal and non-fecal species, suggesting that fecal strings influence bacteria interactions at the egg stage.

A total of 2,229 connections had significant Spearman rank correlations (*P*adjust < 0.01, rs > 0.25 or rs < −0.25) among microbes at egg stages. The strongest positive correlation at the egg stage was between *Ciceribacter* and *Yersinia* (rs = 1.0), while the

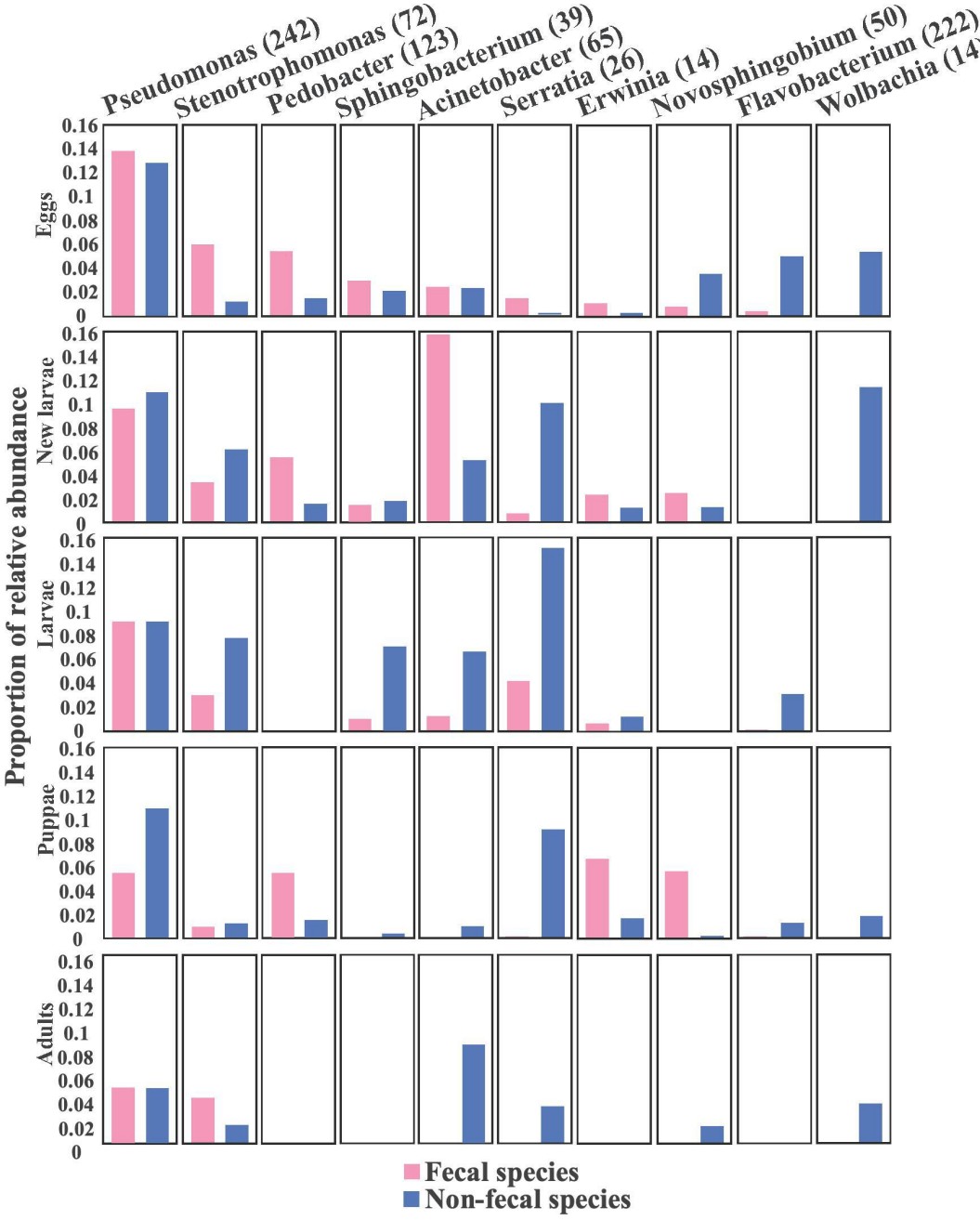

**FIG 2** Relative abundances of the 10 most abundant bacteria genera within fecal and non-fecal beetle species. Numbers in the brackets show ASVs per genus.

strongest negative correlation was between *Kytococcus* and *Pseudomonas* ($rs = -0.50$). Specifically, 48, 12, and 13 correlations were detected between *Wolbachia*, *Pseudomonas*, and *Stenotrophomonas* with other bacteria, respectively. *Wolbachia* and *Flavobacterium* were identified as hub bacteria in non-fecal species at the egg stage and interacted frequently with other bacteria in these species. In contrast, *Pseudomonas*, *Stenotrophomonas*, *Pedobacter*, *Sphingobacterium*, *Acinetobacter*, *Serratia*, *Erwinia*, and *Novosphingobium* interacted more frequently with each other.

## DISCUSSION

The analysis of temporal dynamics of the gut microbial community structure across *Galerucella* life stages revealed a community that is dynamic depending on whether

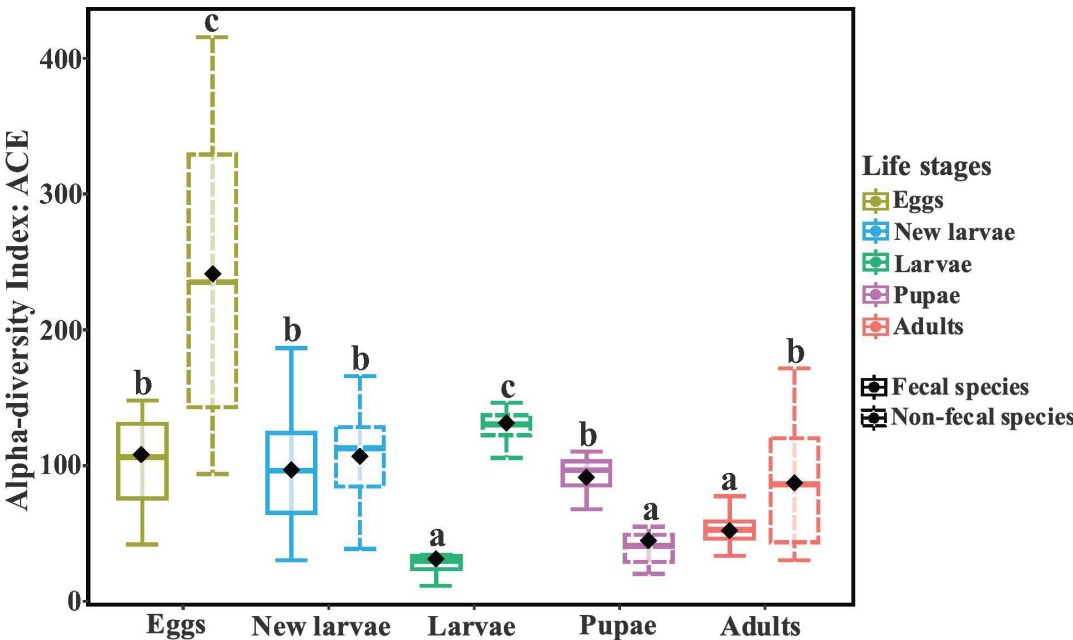

**FIG 3** ACE index of fecal and non-fecal species. We used the ACE diversity measure to account for observed ASVs and estimate unobserved species based on low-abundance ASVs (richness). Different letters (a, b, c) indicate statistically significant differences (p < 0.05).

females deposited a fecal string on the eggs. Microbial diversity was relatively stable between life stages in fecal species but much more variable in non-fecal species. Specifically, the diversity of dominant gut bacteria taxa in non-fecal species peaked during the egg stage, and then decreased significantly from the egg to the pupal stages (Fig. 3). We also observed substantial differences in microbial communities between beetle species, as reported previously (10), and identified several bacteria that consistently appeared across life stages and that differed between beetle hosts. For instance, *Acinetobacter*, *Erwinia*, and *Wolbachia* were exclusively found in non-fecal species, whereas *Novosphingobium* and *Flavobacterium* were more commonly found in fecal species (Fig. 4).

Although the microbial community composition may be strongly affected by metamorphosis in holometabolous insects (20, 21), we found multiple ASVs shared across life stages in both fecal and non-fecal species, indicating potential generational inheritance (Fig. 4). These findings suggest vertical transmission mechanisms in both fecal and non-fecal species, even though non-fecal species lack an apparent transfer mechanism. In fecal species, fecal strings laid on the eggs would efficiently transfer microbial communities from mother to offspring, as these fecal strings are consumed by the newly hatched larvae. Presumably, in non-fecal species, microbes are transferred by females through other mechanisms, such as coating the egg during the egg-laying process (cf. reference 3), although this mechanism appears less efficient and likely reduces the dominance of transferred taxa relative to environmentally acquired taxa. It is likely that fecal strings contain a high number of microbial cells, which presumably results in a high dominance of transferred taxa in fecal species and a more stable microbial dynamic.

The colonization of gut microbial communities in offspring is crucial and affects internal dynamics. This process can be affected by secretions that parent insects add on the eggs (known as pre-hatch care) or by females feeding offspring with bacteria (known as full care) (21). For instance, studies on burying beetles have shown that larvae receiving pre-hatch and full care are colonized with bacteria from the maternal gut, whereas larvae that do not receive such care acquire bacteria from the carcass (21). Similarly, in the oriental fruit fly (*Bactrocera dorsalis*), bacteria (*Citrobacter freundii*) were found in deposited eggs despite not being detected in ovaries (22). Additionally, in

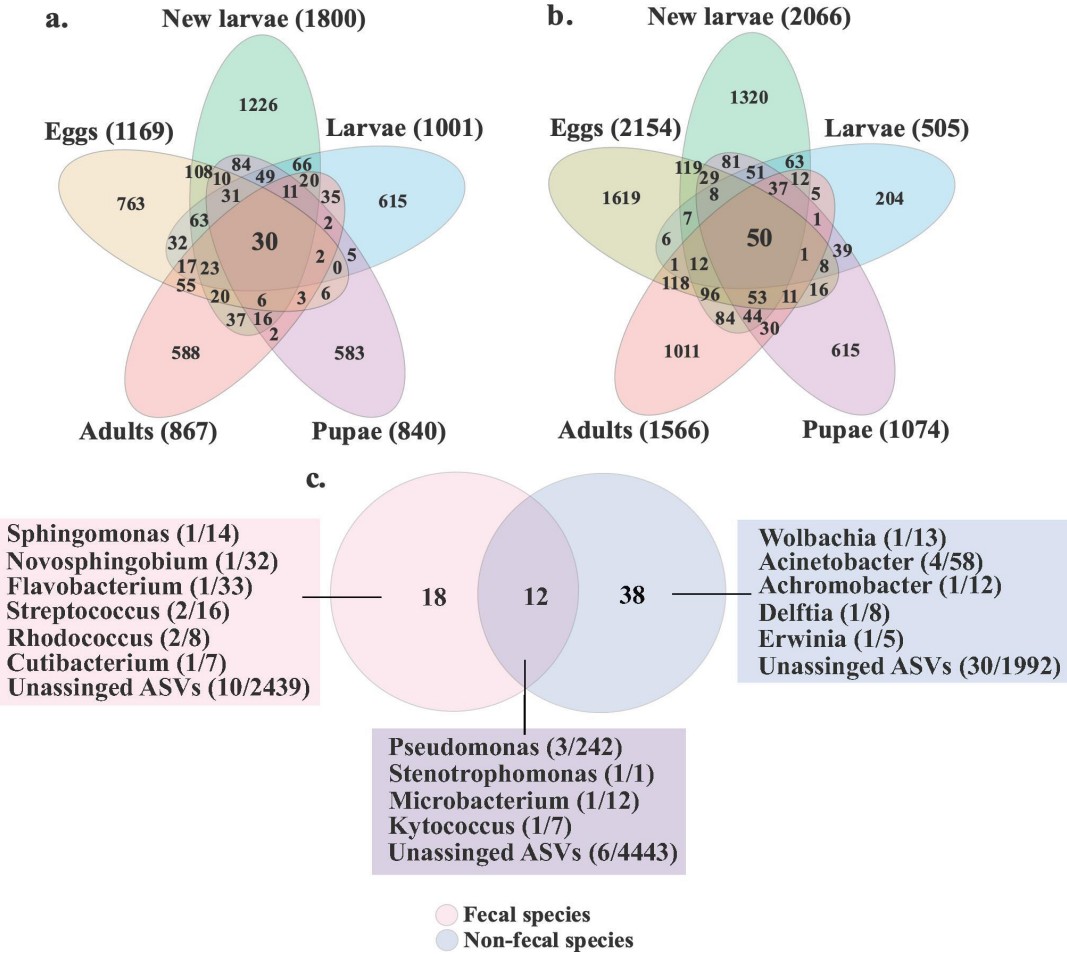

**FIG 4** Venn diagram of ASV distribution across *Galerucella* life stages. Numbers within compartments indicate ASV counts of in (a) fecal species, (b) non-fecal species, and (c) those that are shared across life stages. Numbers in brackets show the total ASV counts (a, b) and the count of shared ASVs/total count for respective group (fecal or non-fecal species) (c).

bumblebees, core gut bacteria persist throughout different development stages, and observed similarities in microbial communities between generations suggest vertical transmission of gut bacteria (20). These examples indicate that endosymbionts could be vertically transmitted through mechanisms, such as smearing, similar to fecal strings. Transmission mode, especially vertical transmission, can also influence host dependence, reproduction, and influence niche colonization (23, 24).

In other species, symbionts can be transferred through elaborate mechanisms within the host reproductive system. For instance, in the rice weevil (*Sitophilus oryzae*), principal symbionts are associated with both primordial germ cells and the future bacteriome and transferred to the next generation through the ovaries (25). There may even be pseudo-vertical transmission, as is the case between the entomopathogenic nematode *Steinernema carpocapsae* and its symbiont, *Xenorhabdus nematophila*. In this case, symbionts are transmitted not only from the parent (vertical) but also from co-occurring nematodes in their host (26). Additionally, the symbionts of *Drosophila melanogaster* are found in the male germ line (27), illustrating the diversity of vertical transmission routes among species. We propose that fecal strings represent an additional transfer mechanism that has received limited attention in insect research. Our understanding of transmission mechanisms does not yet encompass their causes and consequences. However, studying the dynamics of microbial communities between fecal and non-fecal species could highlight crucial factors related to the mode of transmission.

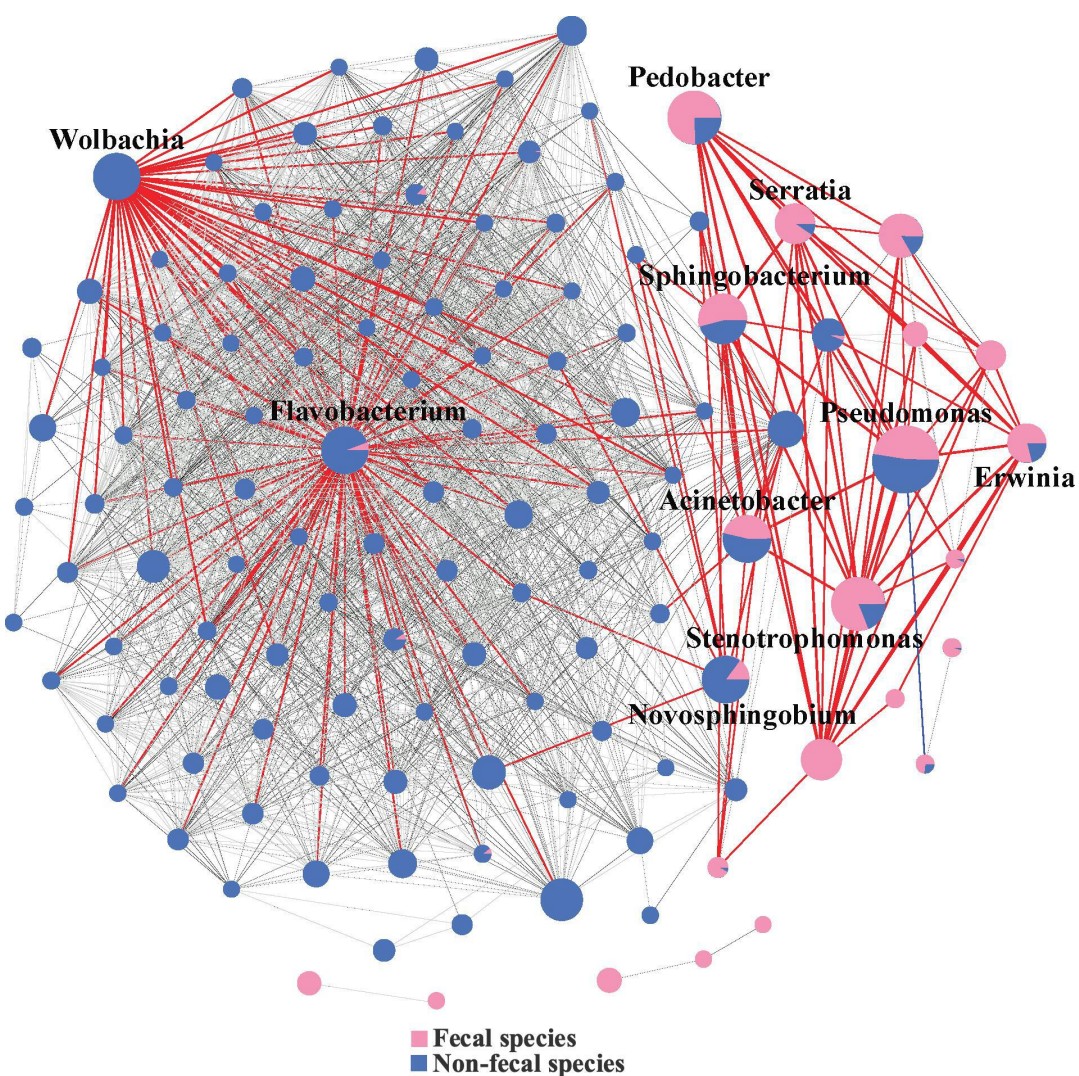

**Fecal species**
**Non-fecal species**

**FIG 5** Co-occurrence bacterial network with all significant connections (*P*adjust < 0.01, rs > 0.25, or rs < −0.25) among bacterial genera at egg stage based on Spearman's rank correlation coefficient. Network nodes represent core bacterial genera, and edges indicate significant correlations between nodes. The size of the nodes reflects the degree of connection, and the thickness of the edges indicates the strength of the correlation. Dot color represents the relationship between relative abundance of bacterial genera and breed composition. Red lines indicate positive co-occurrence patterns, and blue lines indicate negative co-occurrence patterns among the top 10 ASVs.

The microbial diversity in the beetle guts was notably high. ASVs belonging to genera, such as *Pseudomonas*, *Stenotrophomonas*, *Sphingobacterium*, *Acinetobacter*, *Serratia*, *Erwinia*, *Novosphingobium*, *Flavobacterium*, and *Wolbachia* were dominant across different life stages of the *Galerucella* species (Fig. 2 and 4), though their prevalence varied among species. The effects of these core bacteria and how differences in the microbial community differentially impact species remain speculative. In some cases, functions are known from other insect taxa, and the common bacteria in *Galerucella* include both beneficial symbionts and potentially pathogenic species. For instance, *Serratia* dominated non-fecal species across life stages but was less prevalent in fecal species (Fig. 2). This genus is known to contain both gut symbionts found in many insect species (28–32) and species that are insect pathogens (33). Other common symbionts in this study, such as *Stenotrophomonas* and *Pseudomonas*, have been previously shown to depress plant defenses (34, 35), whereas Acinetobacter is known to degrade complex organic molecules (36). In contrast, the functions of some bacteria are less clear. For instance, *Erwinia*, which was commonly found in non-fecal species, is known as a

phytobacterial pathogen that causes fire blight and typically infects plants through insects, such as aphids (37, 38).

In our study, *Wolbachia* was found only in non-fecal species and is a well-known facultative symbiont in insects. These bacteria are often maternally inherited and associated with feminization, parthenogenesis, and cytoplasmic incompatibility (39, 40). For *Wolbachia* to become established in a new host species, it must be horizontally acquired and vertically transmitted at a high rate (41). In *D. mauritiana*, *Wolbachia* may transfer between populations through three possible mechanisms: as neutral bacteria, as beneficial bacteria, or in association with the spindle apparatus during cell division (42). Studies on mites (Acari) have shown that *Wolbachia*-infected females can transfer *Wolbachia* to offspring, regardless of the infection status of males, while offspring of *Wolbachia*-free females remain uninfected (43). In *Galerucella* species, we observed that non-fecal species have a much higher level of *Wolbachia* compared to fecal species, often almost exclusively (Fig. 2 and 5). This pattern could suggest that fecal strings might hinder *Wolbachia* transmission or establishment potentially due to the maintenance of other beneficial microorganisms.

The microbiome dynamics may also be affected by host traits, including diet breadth and type of diet. For instance, species with a broader diet may be exposed to a broader set of environmentally acquired bacteria and thereby show larger variability. Not surprisingly, it is frequently observed that diet is an important determinant of the gut microbiome (e.g., 44–46), but it is yet unclear whether this variability is due to microbes picked up from the plant or adaptations in the microbiome to different diets. So far, few studies have explored microbial community dynamics across life stages in any species, which makes it difficult to formulate general conclusions. Besides our study, only one study known to us has quantified alpha diversity of gut bacteria across life stages (47). In that study involving a polyphagous bug, alpha diversity was highest in eggs, similar to the non-fecal species in this study, and then decreased in later life stages. Our system mainly contains species that are functionally monophagous, even though *G. lineola* may shift among *Salix* species. We hypothesized a higher variability in gut microbiomes of polyphagous species. Further studies are clearly needed to test the hypothesis that the gut microbiome dynamics vary with diet breadth.

To conclude, our findings demonstrate that the gut bacteria dynamics and the bacterial community are influenced by fecal strings across host life stages, revealing distinct microbiome in fecal and non-fecal species. Importantly, this study suggests that different bacterial genera may be either positively or negatively related to fecal strings. Understanding the temporal dynamics of microbial communities in both fecal and non-fecal species could help identify key factors driving microbiome assembly and elucidate how microbial functions relate to insect fitness. Furthermore, we observed that the gut microbiota is shaped by both phylogenetic and environmental factors, where the innovation of adding fecal strings to the egg could play a pivotal role in this particular beetle genus.

## ACKNOWLEDGMENTS

We are thankful for the sequencing supported through the National Genomic Infrastructure (NGI) and Uppmax through the Science for Life Laboratory, which is funded by the Knut and Alice Wallenberg Foundation and the Swedish Research Council Vetenskapsrådet.

This study was possible through a grant from The Swedish Research Council Vetenskapsrådet to PAH (#2019-4980).

P.A.H. held the funding. P.A.H. and Y.A. designed the research. Y.A. performed the research. Y.A. analyzed and visualized the data. Y.A. wrote the initial paper. P.A.H. and S.L.G. supervised the study. P.A.H., S.L.G., and Y.A. read and approved the final version of the manuscript.

## AUTHOR AFFILIATIONS

[1]Department of Ecology, Environment and Plant Sciences, Stockholm University, Stockholm, Stockholm County, Sweden

[2]Institute for Chemistry and Biology of the Marine Environment (ICBM), Carl von Ossietzky Universität Oldenburg, Oldenburg, Lower Saxony, Germany

## AUTHOR ORCIDs

Yueqing An http://orcid.org/0000-0003-0267-7106
Sarahi L. Garcia http://orcid.org/0000-0002-8622-0308
Peter A. Hambäck http://orcid.org/0000-0001-6362-6199

## FUNDING

| Funder | Grant(s) | Author(s) |
| --- | --- | --- |
| Vetenskapsrådet | #2019-4980 | Peter A. Hambäck |

## AUTHOR CONTRIBUTIONS

Yueqing An, Conceptualization, Data curation, Formal analysis, Investigation, Methodology, Project administration, Software, Validation, Visualization, Writing – original draft, Writing – review and editing | Sarahi L. Garcia, Supervision, Writing – review and editing | Peter A. Hambäck, Conceptualization, Funding acquisition, Project administration, Resources, Supervision, Writing – review and editing

## DATA AVAILABILITY

The data that support the findings of this study are openly available in Dryad at 10.5061/dryad.m63xsj49s.

## ADDITIONAL FILES

The following material is available online.

### Supplemental Material

**Supplemental figures (mSystems01723-24-S0001.docx).** Fig. S1 to S4.
**Legends (mSystems01723-24-S0002.docx).** Supplemental figure legends.

### Open Peer Review

**PEER REVIEW HISTORY (review-history.pdf).** An accounting of the reviewer comments and feedback.

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
