## [Reviewer comments · mSystems]

Microbial transfer through fecal strings on eggs affects leaf beetle microbiome dynamics

Yueqing An, Sarahi Garcia, and Peter Hambäck

Corresponding Author(s): Yueqing An, Stockholms Universitet

Review Timeline:

Submission Date:	January 11, 2025
Editorial Decision:	March 3, 2025
Revision Received:	March 18, 2025
Accepted:	April 10, 2025

Editor: Lucy Moleleki

Reviewer(s): The reviewers have opted to remain anonymous.

Transaction Report:

DOI: <https://doi.org/10.1128/msystems.01723-24>

Re: mSystems01723-24 (Microbial transfer through fecal strings on eggs affects leaf beetle microbiome dynamics)

Dear Dr. Yueqing An:

Thank you for the privilege of reviewing your work. Below you will find the reviewer comments to be addressed.

Please return the manuscript within 30 days; if you cannot complete the modification within this time period, please contact me. If you do not wish to modify the manuscript and prefer to submit it to another journal, notify me immediately so that the manuscript may be formally withdrawn from consideration by mSystems.

Revision Guidelines

Sincerely,
Lucy Moleleki
Editor
mSystems

Reviewer #2 (Comments for the Author):

The authors have addressed some of my concerns while leaving others unresolved. I would like the authors to further discuss the impact of diet breadth, specifically, generalist (G1), intermediate (Gs), and specialist (Gp, Gc, Gn) feeding strategies, on the vertical transfer of gut microbes across insect generations, with or without the presence of a fecal string. The authors may speculate or propose a hypothesis based on the current dataset.

REVIEWER REPORTS

Reviewer #2 (Comments for the Author):

The authors have addressed some of my concerns while leaving others unresolved. I would like the authors to further discuss the impact of diet breadth, specifically, generalist (G1), intermediate (Gs), and specialist (Gp, Gc, Gn) feeding strategies, on the vertical transfer of gut microbes across insect generations, with or without the presence of a fecal string. The authors may speculate or propose a hypothesis based on the current dataset.

RESPONSE: We have now added a paragraph in the discussion towards this end, and hope that this is satisfactory.

Re: mSystems01723-24R1 (Microbial transfer through fecal strings on eggs affects leaf beetle microbiome dynamics)

Dear Dr. Yueqing An:

Your manuscript has been accepted, and I am forwarding it to the ASM production staff for publication. Your paper will first be checked to make sure all elements meet the technical requirements. ASM staff will contact you if anything needs to be revised before copyediting and production can begin. Otherwise, you will be notified when your proofs are ready to be viewed.

Sincerely,
Lucy Moleleki
Editor
mSystems